# Exploring Embryo-Ototoxic Effects: Insights into Deodorant-Induced Hair Cell Damage in Zebrafish

**DOI:** 10.3390/ijms25020948

**Published:** 2024-01-12

**Authors:** Hee Soo Yoon, Kyung Tae Hyun, Sumin Hong, Saemi Park, Eunjung Han, Hyun woo Baek, Yun Kyoung Lee, Kang Hyeon Lim, Yoon Chan Rah, June Choi

**Affiliations:** 1Department of Otorhinolaryngology-Head and Neck Surgery, Korea University College of Medicine, Ansan Hospital, Ansan 15355, Republic of Korea; yhsjoa@hanmail.net (H.S.Y.); vlalr@naver.com (K.T.H.); hongsumin0608@naver.com (S.H.); babydazzler@gmail.com (S.P.); hoj7843@naver.com (E.H.); bhw0810@naver.com (H.w.B.); leeyk83@gmail.com (Y.K.L.); kingsonl@hanmail.net (K.H.L.); ycrah@naver.com (Y.C.R.); 2Biomedical Research Center, Korea University College of Medicine, Ansan Hospital, Ansan 15355, Republic of Korea; 3Zebrafish Translational Medical Research Center, Korea University, Ansan 15355, Republic of Korea

**Keywords:** zebrafish, ototoxicity, hair cell, audiologic result, behavior change

## Abstract

Our study investigated the embryo-ototoxic effects of deodorant2 (DA2) on zebrafish embryos, which serve as valuable model organisms due to genetic and physiological similarities to humans. We focused on understanding DA2’s impact on zebrafish hair cells, which are vital for sensory perception and balance regulation. DA2, provided by the Ministry of Environment, Republic of Korea, was used at 460 μg/mL in dimethyl sulfoxide (DMSO), with a 0.43% DMSO solvent control group. Three experiments, each using 10 zebrafish specimens from each group, showed an initial 13% hair cell count reduction in the DA2-exposed group. Subsequent experiments demonstrated reductions of 37% and 22%, each with one mortality case. Statistical analysis revealed a significant 24% hair cell count reduction in the DA2-exposed group. We also assessed DA2’s impact on zebrafish behavior. Although not statistically significant, differences in distances traveled (0.33–0.39, 95% confidence interval: −0.46–1.1, *p* = 0.2033) and latencies (−0.016–0.018, 95% confidence interval: −0.052–0.021, *p* = 0.1917) hinted at negative effects. These results highlight DA2’s ototoxic properties affecting zebrafish auditory systems and behavior. Further investigation into DA2’s effects on aquatic organisms and potential mitigation strategies are essential. These findings contribute to understanding DA2’s safety profile, benefiting aquatic ecosystems and human health assessments.

## 1. Introduction

We aimed to investigate the embryo-ototoxic effects of a substance known as deodorant2 (DA2), a name randomly assigned to one of several deodorants previously tested in an experiment on zebrafish. DA2 is a commercially available deodorant that has been studied for its ototoxicity. The toxicity of deodorant ingredients is primarily influenced by their chemical composition and the specific type of deodorant products dominating the market [1]. Recent studies have demonstrated the developmental toxicity of deodorants. Well-known compounds found in deodorants, such as triclosan (TCS), share structural similarities with endocrine disruptors, steroids, and thyroid hormones, leading to hair cell damage [2]. Solvents used in deodorants, such as toluene and other forms of volatile organic compounds, are also known to cause ototoxicity [3,4,5]. These environmental factors may cause hearing loss by damaging hair cells in living organisms [6]. The risks of deodorant usage have been reviewed for various organs, including the liver, skeletal muscle, endocrine system, and other metabolic processes [7,8,9,10,11,12]. However, there is insufficient research regarding the effects of deodorants on hair cell damage or ototoxicity. Further experiments are required to analyze the possibility of DA2-induced hair cell damage. Therefore, we used zebrafish, a useful model for hair cell research. Numerous benefits are associated with the use of zebrafish as a model organism [13], including economic upkeep, a swift life cycle, prolific reproduction yielding substantial offspring, and externally observable transparency during embryonic development [14]. A single pair of mating zebrafish can yield several hundred progenies, which undergo expedited embryonic transformation from eggs to self-sustaining, motile larvae within a span of five days. This combination of attributes makes zebrafish an important vertebrate model for genetic and behavioral investigations [15].

Embryos of zebrafish, owing to their striking physiological and genetic resemblance to humans, have emerged as valuable model organisms for toxicological studies [16]. The effect of many toxicants on zebrafish embryos is well correlated with those observed in rodents [17]. This study focused on unraveling the potential adverse effects of DA2 on the delicate hair cells of zebrafish, which are the crucial components responsible for sensory perception and balance regulation [18]. Conventionally, live hair cells are easily visualized in vivo in optically clear embryos via staining with 2-(4-(dimethylamino)styryl)-N-ethylpyridinium iodide, a fluorescent styryl dye [19]. The advantages of zebrafish’s optical clarity have become evident in the efficient analysis of the morphology and functionality of hair cells after drug treatment. In addition, zebrafish exhibit a wide array of motor behaviors that are neurologically initiated by their sensory organs, such as the lateral line or auditory system [20]. Their startle response exhibits distinct and consistent characteristics, readily activated by a simple tap on the zebrafish enclosure. Thus, these characteristics enabled us to employ the startle response as a reliable behavioral tool for evaluating hair cell damage and associated intervening factors [21,22].

The results of this study provide insights into the ototoxic properties of DA2 and its implications for auditory health. Understanding the risks associated with DA2 exposure in zebrafish can contribute to a broader understanding of its safety profile, thereby benefiting human health assessments. In addition, it contributes to a more comprehensive understanding of the safety profile of the material.

## 2. Results

The mean number of hair cells within the four neuromasts (SO1, SO2, O1, and OC1) on one side of each fish was counted under a fluorescence microscope to quantitatively assess changes.

Three experiments were conducted, each using 10 zebrafish specimens from each group. The experimental design included a control group that did not undergo any experimental intervention, a solvent control group to evaluate solvent toxicity, and an experimental group treated with DA2. Following the experimental procedures outlined in the method section, zebrafish specimens were processed immediately, and the number of hair cells in each group was quantified to assess the toxicity of DA2.

In the initial experiment, the DA2-exposed group exhibited a significant 13% reduction in hair cell count compared with the control group, indicating substantial damage to auditory structures. In the subsequent experiment, the DA2 group showed a 37% decrease in hair cell count relative to that of the control group, with one mortality. Finally, in the third experiment, one specimen also suffered mortality, and the hair cell count exhibited a substantial decrease of 22% compared with the control group (Figure 1).

Collectively, a comprehensive statistical analysis of the three experiments, each comprising a sample size of 30 individuals per each group, showed that exposure to DA2 induced a significant 24% reduction in hair cell count compared with the control group (Figure 2).

In this study, we conducted an experiment to investigate the effects of DA2 on zebrafish behavior. The behavioral experiment comprised a control group and a group exposed to DA2. The distances traveled by each group were recorded over a specific timeframe, and the latency of their movements was noted (Figure 3).

In our behavioral experiments, entities with a latency exceeding 0.5 s were considered unresponsive to the stimulus, and were thus treated as missing data in the dataset. The selection of the 0.5 s threshold was determined arbitrarily based on the researchers’ experiential judgment.

Statistical analysis using an independent *t*-test revealed that the range of mean distance differences between groups was 0.33–0.39 mm, with a 95% confidence interval of −0.46–1.1, suggesting no statistically significant differences between the two groups (*p* = 0.2033). In addition, the range of mean latency differences between groups was −0.016–0.018, with a 95% confidence interval of −0.052–0.021, suggesting no statistically significant differences between the two groups (*p* = 0.1917). Despite the absence of statistically significant differences in the results, both groups exhibited a discernible trend in the adverse effects induced by DA2 (Figure 4).

## 3. Discussion

This study highlighted the importance of evaluating the ototoxic effects of various compounds, particularly those with potential environmental impacts. Our findings highlight the necessity of thoroughly screening substances that can influence various biological systems in different environments, thereby emphasizing the need to understand their potential risks. The risk posed by DA2, a deodorant, is a subject of concern because of its involvement in multiple physiological processes, including its known effects on the heart, brain, and liver, as mentioned earlier. However, assessment of its potential harm to specialized sensory cells, such as hair cells, has been limited. In this study, we addressed this gap by examining potential hair cell damage induced in zebrafish by DA2.

Owing to its high sensitivity, zebrafish behavior is frequently employed as a screening tool to evaluate the ototoxic potential of pharmaceutical compounds [23,24,25]. Moreover, the zebrafish model has been instrumental in elucidating and characterizing the roles of genes critical for hair cell synapse function. Notably, at the molecular and cellular levels, zebrafish hair cells exhibit a striking resemblance to their mammalian counterparts [26].

Our study provides valuable insights into the toxicological effects of DA2, specifically focusing on its effects on hair cells. Using a zebrafish model, we assessed the potential ototoxicity of DA2 and elucidated its effects on the sensory organs. The structural similarity of hair cells in zebrafish to those found in the human inner ear renders zebrafish an exemplary model organism for investigating inner ear dysfunction [27,28,29]. This analysis contributes to a better understanding of the overall toxicity profile of DA2 and highlights its potential implications for human health.

A novel aspect of this study is its translational potential. Data obtained from this study not only deepen our knowledge of DA2-induced hair cell damage, but also provide a foundation for future clinical applications. The use of a zebrafish model to screen for potential toxic effects could pave the way for the identification of compounds with harmful properties before they enter clinical trials or before environmental exposure. Historically, substances like cisplatin, gentamicin, quinine, and neomycin have been linked to inducing ototoxicity [30,31,32,33]. Although a direct comparison of concentration ratios for these substances was not performed, it can be inferred that the chemical compounds present in DA2, the deodorant used in this study, may have a comparable potential to elicit ototoxicity when compared to the known ototoxicity rates associated with these substances.

Despite these significant insights, it is important to acknowledge the limitations of this study. Although useful, zebrafish models may not fully replicate the complexity of the human system. Further research should aim to validate our findings in other animal models, and ultimately, in human studies. Additionally, DA2 concentrations used in our experiments may not precisely reflect real-world exposure scenarios, warranting caution when extrapolating our results to real-life situations. Still, through our examination of DA2-induced hair cell damage in zebrafish, we provided valuable insights into the toxicological evaluation of this compound.

## 4. Materials and Methods

### 4.1. Prepared Materials and Zebrafish

The material DA2 was prepared by the Ministry of Environment, Republic of Korea. It is a commercially available material functioning as an odor remover and can be employed in diverse locations. The commercial name of DA2 is “Bullsone Odor Free-Mountain mist” (Bullsone Company, Seoul, Republic of Korea). The components of DA2 are multifaceted, with the predominant constituents, excluding purified water, being ethanol (used as a solvent), surfactants, green tea, herbal extracts, fragrances, and the preservative benzisothiazolinone. The experimental concentration of the material was 460 μg/mL, dissolved in DMSO. For the solvent control group, the concentration was adjusted to 0.43% DMSO to evaluate the effect of the solvent.

Wild-type zebrafish were maintained in the embryo medium (15 mM NaCl, 0.5 mM KCl, 1 mM CaCl_2_, 1 mM MgSO_4_, 0.15 mM KH_2_PO_4_, 0.05 mM NH_2_PO_4_, and 0.7 mM NaHCO_3_) under a regular photoperiod (14 h light:10 h dark). Zebrafish embryos were produced by mating adult fish maintained at 28.5 ± 1 °C in a zebrafish facility at our hospital (Korea University Zebrafish Translational Medical Research Center, Ansan, Republic of Korea). All protocols were conducted in accordance with the guidelines of the Animal Care Ethics Committee of the Korea University Medical Center and National Institutes of Health (Approval No.: KOREA-2018-0054).

After washing the larvae three times with embryo medium, DA2 was added to the embryo medium at a concentration of 460 μg/mL for 120 h. At 120 h post-fertilization, the larvae were rinsed with embryo medium three times and anesthetized using tricaine (3-aminobenzoic acid 0.4 g/ethyl ester; 100 mL; pH 7, adjusted using Tris buffer) for 5 min. A negative control group containing no additional chemicals was also established. The larvae were then mounted on a depression slide using methylcellulose and assessed under a fluorescence microscope. The mean number of hair cells within the four neuromasts (supraorbitals 1 and 2 (SO1 and SO2), otic (O1), and occipital (OC1)) on one side of each fish was counted under the fluorescence microscope. Hair cells in the zebrafish were counted using a technique previously validated in our research laboratory [34,35].

### 4.2. Analysis of Behavioral Changes

Here, we employed a well-established method to assess the startle response in zebrafish larvae, which was crucial for evaluating their behavior. After recording the baseline behavior of larvae for 5 min in a static state, a 10 min interval was allowed to ensure stabilization of the larvae’s activity. Subsequently, larvae were subjected to standardized tapping stimulation using the DanioVision system. To measure the startle response, we focused on two key parameters: latency, which represents the time elapsed from the tapping stimulation to the initiation of the response, and peak distance moved, which quantifies the distance covered by the larvae during the startle response. The methodology for the startle response assessment was adapted from previously established protocols [36,37].

Notably, the startle response was deemed valid only if the larvae exhibited a rapid bout of activity within 0.5 s of the tapping stimulation and moved more than 0.6 mm at the reflex moment. This standardized approach allowed us to consistently measure and compare the startle responses of zebrafish larvae in a controlled and replicable manner, thereby contributing to the overall robustness of the experimental design. A similar procedure was previously conducted in our laboratory [38]. An example recording of the startle response is shown in Appendix A.

### 4.3. Statistical Analysis

All data are presented as mean ± standard deviation. One-way analysis of variance was used for multiple comparisons, with the significance level set at *p* < 0.05. A post-hoc analysis was conducted using Tukey’s honestly significant difference test. Additionally, an independent *t*-test was used to analyze behavioral changes. Statistical analyses were performed using IBM SPSS 20.0 for Windows (IBM, Armonk, NY, USA) and Prism 7 for Windows (La Jolla, CA, USA).

## 5. Conclusions

In conclusion, this study highlighted the embryo-ototoxicity of DA2 and its detrimental effects in zebrafish. This study established a link between DA2 exposure and hair cell damage that led to behavioral changes. These findings contribute to our understanding of the potential risks associated with DA2, and underscore the need for further investigation of its effects on aquatic organisms. Future studies should explore additional mechanisms underlying the observed damage and potential strategies to minimize the adverse effects of DA2 on aquatic ecosystems.

## Figures and Tables

**Figure 1 ijms-25-00948-f001:**
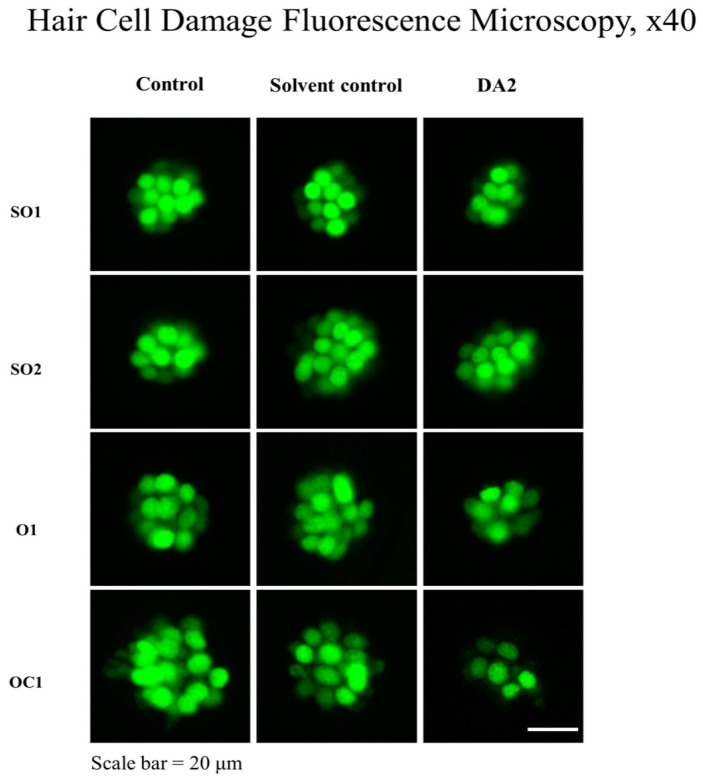
Zebrafish hair cell damage shown via fluorescence microscopy, ×40. Compared with control groups, DA2-exposed groups displayed decreased numbers of hair cells. A significant reduction in total hair cell counts across the four neuromasts was observed in DA2-exposed groups. Scale bar = 20 μm; SO1, supraorbital 1; SO2, supraorbital 2; O1, otic 1; OC1, occipital 1.

**Figure 2 ijms-25-00948-f002:**
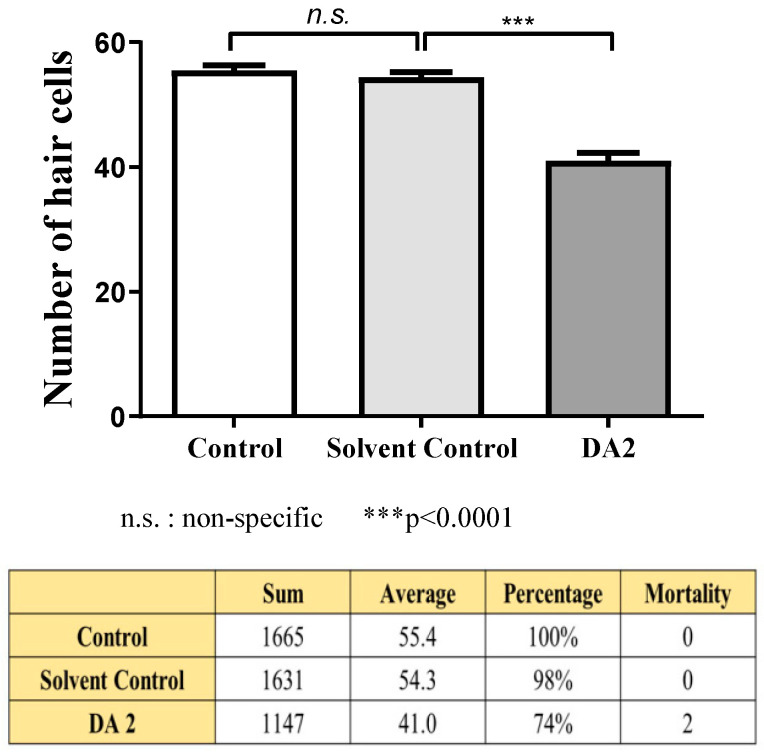
The DA2 group exhibited a significant reduction in hair cell count when compared to both solvent control and control groups, as determined using ANOVA (*p* = 0.0472). Additionally, no mortality was observed in the control group, whereas two fatalities were documented in the DA2 experimental group.

**Figure 3 ijms-25-00948-f003:**
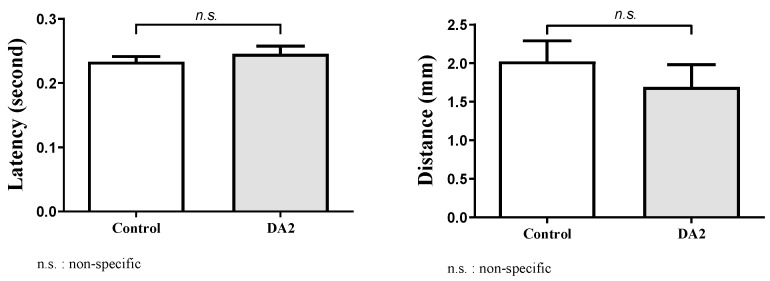
Comparison of latency and distance between DA2-exposed and control groups. Statistical analysis using an independent *t*-test revealed that the range of mean distance differences between groups was 0.33–0.39 mm, with a 95% confidence interval of −0.46–1.1 (*p* = 0.2033). The range of mean latency differences between groups was −0.016–0.018, with a 95% confidence interval of −0.052–0.021 (*p* = 0.1917).

**Figure 4 ijms-25-00948-f004:**
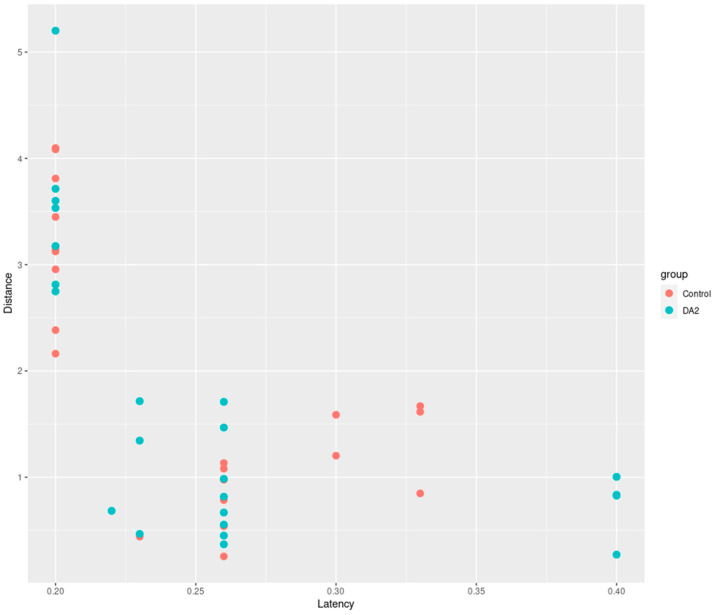
Comparison of latency and distance simultaneously in a single figure between the DA2-exposed and control groups. Although not statistically significant, observed results indicated a trend towards unfavorable effects in both latency and distance when compared with the control group.

## Data Availability

Datasets generated and/or analyzed in the current study are available from the corresponding author upon reasonable request.

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
