# Peer review of "Exploring Embryo-Ototoxic Effects: Insights into Deodorant-Induced Hair Cell Damage in Zebrafish"

_ijms, 2024, doi:10.3390/ijms25020948_

Round 1

Reviewer 1 Report

Comments and Suggestions for Authors

The manuscript chose the model fish species zebrafish as the test organism to explore embryo-ototoxic effects of DA2 based on hair cell damage. The manuscript is somewhat innovative, and the experimental design is reasonable, but the results are not well discussed. Therefore, I believe that MAJOR REVISION is needed before it could be accepted for publication.

Specific Comments:

[1] In the Introduction section, the authors should have given more information on the developmental toxicity of DA2, especially the research progress of ototoxicity studies and the association between hair cell damage and ototoxicity, rather than giving the readers well-known information about the advantages of zebrafish as a model fish species.

[2] Line 31-32, add the references;

[3] Please describe why the authors chose 10 zebrafish specimens as the sample size based on reasonability and accordance.

[4] In the Discussion section, I strongly suggest that the authors re-analyze and re-discuss the reasonability and logicality between the obtained data in this manuscript and other DA2-ototoxicity studies or compare with the findings of other environmental contaminants-induced ototoxicity and hair cell damage. 

[5] Please provide more information on the sources and strains of zebrafish used in this study.

[6] Please describe why the authors chose 1 h as the exposure period. Please give a detailed guideline information.

Comments on the Quality of English Language

No serious linguistic errors. But i suggest the authors re-check the grammar throughout.

Author Response

[1] In the Introduction section, the authors should have given more information on the developmental toxicity of DA2, especially the research progress of ototoxicity studies and the association between hair cell damage and ototoxicity, rather than giving the readers well-known information about the advantages of zebrafish as a model fish species.

: Thank you for the precise review of the content. As suggested, we have added information on the developmental toxicity of DA2, including relevant ototoxicity studies and references. Additionally, a reference highlighting the ototoxic potential of solvents used in deodorant formulation has been incorporated.

[2] Line 31-32, add the references;

: References have been added as requested.

[3] Please describe why the authors chose 10 zebrafish specimens as the sample size based on reasonability and accordance.

: We have conducted similar experiments multiple times in the past, involving exposure studies with fine particulate matter, silver nanoparticles, nicotine, and others. These experiments consistently yielded statistically significant results with 10 zebrafish specimens per trial. Moreover, practical constraints and considerations of the central limit theorem were taken into account, allowing for a normal distribution approximation with 10 specimens per trial in accordance with statistical guidelines.

[4] In the Discussion section, I strongly suggest that the authors re-analyze and re-discuss the reasonability and logicality between the obtained data in this manuscript and other DA2-ototoxicity studies or compare with the findings of other environmental contaminants-induced ototoxicity and hair cell damage. 

: As DA2 is a commercially available deodorant with no existing studies on its ototoxicity, our experiment serves as the first investigation into this aspect. However, in the Discussion section, we have supplemented the analysis by introducing references to ototoxicity studies involving other substances such as cisplatin, gentamicin, quinine, and neomycin, providing a broader context.

 [5] Please provide more information on the sources and strains of zebrafish used in this study.

: Additional information on the sources and strains of zebrafish used in this study has been incorporated into the Materials and Methods section.

[6] Please describe why the authors chose 1 h as the exposure period. Please give detailed guideline information.

 : Upon reviewing the experimental design, we have corrected the exposure period to 120 hours post-fertilization and added this precise information to the Materials and Methods section.

Comments on the Quality of English Language

No serious linguistic errors. But i suggest the authors re-check the grammar throughout

: We have thoroughly re-checked the grammar throughout the entire article, addressing any potential issues.

Reviewer 2 Report

Comments and Suggestions for Authors

The manuscript titled "Exploring Embryo-Ototoxic Effects: Insights into Deodorant-Induced Hair Cell Damage in Zebrafish" by Yoon et al. aims to provide insights into the ototoxic properties of DA2 and its implications for auditory health using zebrafish as a model.

Please provide a detailed information about DA2 in introduction regarding the reason for its development, scope, and frequency of use by humans that prompted you to design this study.

Author Response

Please provide a detailed information about DA2 in introduction regarding the reason for its development, scope, and frequency of use by humans that prompted you to design this study.

: Thank you for the precise review of the content. In response to your feedback, we have enhanced the introduction by incorporating additional information on DA2. Specifically, we have included references and studies related to ototoxicity and hair cell damage associated with deodorants, reinforcing the theoretical basis for the potential impact of deodorants on the human body. Additionally, we introduced reference highlighting the ototoxic potential of the solvent used in deodorant formulation. This supplementary information strengthens the theoretical foundation regarding the possible effects of deodorants on the human body. We appreciate your valuable feedback and have taken steps to provide a more comprehensive context for the rationale behind our study.

Reviewer 3 Report

Comments and Suggestions for Authors

The work I received for review presents very interesting results put in a quite good form of a scientific publication. However, there are all positive features of this article, which I can see in it. There are many more negative ones. The most important of them should be mentioned here:

- Very little experimental data are presented here. This type of work, even as a communication, does not qualify for a highly rated journal such as IJMS.

- Such work may apply for publication in a journal that has the word "communications" in its title, and preferably it should be a magazine from the cosmetology subject matter.

- In almost the entire work, the tested product is hidden under the meaningless symbol DA2, and only at the end, in lines 166-169, it is explained in more detail form. But this list, in its form , is not sufficient to draw appropriately detailed conclusions.

- It would be much more beneficial to try to determine which ingredient of the approved preparation has the described negative effect.

- Based on the results obtained, the use of DA2 should be very strictly restricted and perhaps even suspended. It is strange the least to say, that a preparation which is in common use has only now been found to have toxic properties, and in a rather trivial test that its manufacturer may have also conducted in one form or another.

- The authors base their conclusions on 3 series of studies only, with a scatter of results of 13-37%; they take the arithmetic mean from it and consider it as the key reference value. In this situation, it is very difficult to talk about the repeatability of the results.

- Practically, the key results for the work are the table included in Fig. 2 and diagram in Fig. 3. The rest is "supplementary" data. This is definitely too little, even for a Communication.

- It's good that the authors themselves can see the shortcomings of their work, which they write about in lines 153-160.

Taking into account the above arguments, I suggest rejecting this work and not considering the possibility of resubmitting it.

Author Response

- In almost the entire work, the tested product is hidden under the meaningless symbol DA2, and only at the end, in lines 166-169, it is explained in more detail form. But this list, in its form, is not sufficient to draw appropriately detailed conclusions.

: First of all, thank you for the precise review of the content. As suggested, we have added information related to ototoxicity and hair cell damage associated with DA2 and deodorant, along with relevant references. Additionally, we included reference (3) highlighting the ototoxic potential of the solvent used in deodorant formulation. This reinforcement provides a theoretical foundation for the potential impact of deodorants on the human body.

- It would be much more beneficial to try to determine which ingredient of the approved preparation has the described negative effect.

: This experiment evaluated the toxicity of a product containing multiple chemicals, rather than a single compound among known substances. Through this approach, we aimed to assess the toxicity of substances commonly encountered in daily life, emphasizing the toxicity of everyday materials rather than individual compounds.

- Based on the results obtained, the use of DA2 should be very strictly restricted and perhaps even suspended. It is strange the least to say, that a preparation which is in common use has only now been found to have toxic properties, and in a rather trivial test that its manufacturer may have also conducted in one form or another.

: Firstly, this could be a limitation of animal experiments. The significant difference in size between humans and zebrafish, coupled with the fact that the experiment began with zebrafish embryos, cannot be ignored. Furthermore, the experiment involved applying the substance to embryos, leading to potential differences in exposure time and quantity compared to typical human exposure to deodorant. Nevertheless, evidence of ototoxicity in animal experiments may warrant further research into the environmental impact on humans in the future.

 The authors base their conclusions on 3 series of studies only, with a scatter of results of 13-37%; they take the arithmetic mean from it and consider it as the key reference value. In this situation, it is very difficult to talk about the repeatability of the results.

: While it may seem that the results are based on a limited number of studies, considering the statistically significant differences demonstrated through statistical analysis, it would be inappropriate to suggest a lack of repeatability. Even with only three studies, the evidence indicates meaningful differences that support our conclusions.

- Practically, the key results for the work are the table included in Fig. 2 and diagram in Fig. 3. The rest is "supplementary" data. This is definitely too little, even for a Communication.

: While the overall size of the dataset may seem limited, the clear statistical confirmation of a significant reduction in hair cells, combined with the integrated results of the concurrently conducted behavior test, lays the groundwork for future, more extensive and advanced research. These key results, presented in the table (Fig. 2) and diagram (Fig. 3), provide a solid foundation for our study.

Round 2

Reviewer 1 Report

Comments and Suggestions for Authors

Thank you for the authors' efforts.

Reviewer 3 Report

Comments and Suggestions for Authors

The following opinion is a second review of the same work after some corrections were made by the authors. These corrections are undoubtedly important for the entire work and improve its quality, but they do not expand the scope of experimental work carried out and do not bring any new knowledge to this work. In this situation, the Editorial Office has two options: either accept the work in its current form or reject it definitively. Taking into account the opinions of other reviewers expressed already in the first round of evaluation, my vote remains in the minority. For this reason, I consent, although without much conviction, to accept this work for publication as a Communication in IJMS.